# The impact of vaccine-linked chemotherapy on liver health in a mouse model of chronic *Trypanosoma cruzi* infection

Duc Minh Nguyen[1], Cristina Poveda[2], Jeroen Pollet[2], Fabian Gusovsky[4], Maria Elena Bottazzi[2,3,5], Peter J. Hotez[2,3,5,6,7], Kathryn Marie Jones[2,3]*

1 Center for Comparative Medicine, Baylor College of Medicine, Houston, Texas, United States of America, 2 Texas Children's Hospital Center for Vaccine Development, Department of Pediatrics, Division of Tropical Medicine, Baylor College of Medicine, Houston, Texas, United States of America, 3 Department of Molecular Virology and Microbiology, Baylor College of Medicine, Houston, Texas, United States of America, 4 Global Health Research, Eisai, Inc., Cambridge, Massachusetts, United States of America, 5 Department of Biology, Baylor University, Waco, Texas, United States of America, 6 James A. Baker III Institute for Public Policy, Rice University, Houston, Texas, United States of America, 7 Hagler Institute for Advanced Study at Texas A&M University, College Station, Texas, United States of America

* kathrynj@bcm.edu

**Data Availability Statement:** All relevant data are within the manuscript.

## Abstract

### Background

Chagas disease, chronic infection with *Trypanosoma cruzi*, mainly manifests as cardiac disease. However, the liver is important for both controlling parasite burdens and metabolizing drugs. Notably, high doses of anti-parasitic drug benznidazole (BNZ) causes liver damage. We previously showed that combining low dose BNZ with a prototype therapeutic vaccine is a dose sparing strategy that effectively reduced *T. cruzi* induced cardiac damage. However, the impact of this treatment on liver health is unknown. Therefore, we evaluated several markers of liver health after treatment with low dose BNZ plus the vaccine therapy in comparison to a curative dose of BNZ.

### Methodology

Female BALB/c mice were infected with a bioluminescent *T. cruzi* H1 clone for approximately 70 days, then randomly divided into groups of 15 mice each. Mice were treated with a 25mg/kg BNZ, 25µg Tc24-C4 protein/ 5µg E6020-SE (Vaccine), 25mg/kg BNZ followed by vaccine, or 100mg/kg BNZ (curative dose). At study endpoints we evaluated hepatomegaly, parasite burden by quantitative PCR, cellular infiltration by histology, and expression of B-cell translocation gene 2(BTG2) and Peroxisome proliferator-activated receptor alpha (PPARα) by RT-PCR. Levels of alanine transaminase (ALT), aspartate transaminase (AST), alkaline phosphatase (ALP) and lactate dehydrogenase (LDH) were quantified from serum.

### Results

Curative BNZ treatment significantly reduced hepatomegaly, liver parasite burdens, and the quantity of cellular infiltrate, but significantly elevated serum levels of ALT, AST, and LDH. Low BNZ plus vaccine did not significantly affect hepatomegaly, parasite burdens or the

**Funding:** This work was funded by a grant from the Southern Star Medical Research Institute to PJH. The Pathology and Histology Core at Baylor College of Medicine supported this work with funding from the NIH (P30 CA125123). The funders had no role in study design, data collection and analysis, decision to publish, or preparation of the manuscript.

**Competing interests:** All authors of this manuscript currently are involved in a Chagas vaccine development program. MEB and PJH are listed among the inventors on a Chagas disease vaccine patent, submitted by Baylor College of Medicine. FG is employed at Eisai Inc.

quantity of cellular infiltrate, but only elevated ALT and AST. Low dose BNZ significantly decreased expression of both BTG2 and PPARα, and curative BNZ reduced expression of BTG2 while low BNZ plus vaccine had no impact.

## Conclusions

These data confirm toxicity associated with curative doses of BNZ and suggest that while dose sparing low BNZ plus vaccine treatment does not reduce parasite burdens, it better preserves liver health.

## Author summary

Chagas disease is a neglected tropical disease caused by the protozoal parasite *Trypanosoma cruzi*, which has long-term deleterious health effects. The current treatment for Chagas disease is administering the antiparasitic drug, benznidazole. While benznidazole effectively treats the disease during the acute phase, its efficacy is reduced during chronic infection. In addition, benznidazole therapy causes significant side effects, including liver toxicity. Texas Children's Hospital Center for Vaccine Development at Baylor College of Medicine has developed a treatment strategy that combines a prototype therapeutic vaccine with a lower dose of Benznidazole to promote a protective immune response, ameliorate the deleterious effects of the parasite, and limit the harmful side effect of the drug. We call this vaccine-linked chemotherapy, which has shown promising results regarding heart health by reducing parasite burden and pathology in the heart and improving cardiac function. This study evaluated the strategy's effectiveness in the liver since it is the prime metabolizer of the benznidazole drug, as well as the organ of parasite clearance. Results from this study demonstrated that vaccine-linked chemotherapy causes less damage to the liver compared to curative doses of benznidazole and may be a desirable treatment strategy to preserve overall health while retaining efficacy.

## Introduction

Chagas disease is a bloodborne parasitic protozoal disease caused by *Trypanosoma cruzi*, which through human migration, is found in all parts of the world [1]. Considered a neglected tropical disease, it is mainly found in lower socioeconomic areas of Latin America, where an estimated 6–7 million people are affected by this disease [2]. Being a bloodborne infection, the disease can be disseminated through blood transmission, organ transplant, or congenital transmission from mother to fetus [3]. However, the most common path of infections is through contact with the feces of infected Triatomine insect vectors, which are only found in the Americas [4].

Most often, people are not even aware of their infection status. In the initial acute phase of infection, the infected individual may be asymptomatic or experience nonspecific, generalized flu-like symptoms such as fever, fatigue, body aches, vomiting, and diarrhea [1,5]. During this time, high levels of motile trypomastigotes circulate throughout the individual's bloodstream [6]. Left untreated, the disease progresses to a chronic phase, circulating parasite levels in the blood drop to low levels, and detection can be difficult without sensitive PCR or culture methods [7,8]. There are also long-term deleterious health effects associated with the chronic phase. 30% of people develop cardiac complications, which is the most significant disease manifestation and may include heart enlargement, conduction disturbances, or even cardiac arrest

[9,10]. In 10% of people, gastrointestinal complications may occur, causing disorders like megaesophagus or megacolon [11].

Though cardiac disease, and to a lesser extent gastrointestinal disease, have been extensively explored, the impact of *T. cruzi* on the liver need further elucidation. Studies have shown that the liver is important in clearing parasites from the blood during both the acute and chronic phases of infection [12]. The release of amastigote nest and immediate phagocytosis by resident immune cells produce nitric oxide and oxygen radicals that kill the parasite but can lead to organ damage [13]. Once the disease enters the chronic phase, it is characterized by low parasitemia and low-grade tissue inflammation [7]. Reactive oxygen species (ROS) can directly stimulate hepatic stellate cells, which are known to produce extracellular matrix proteins that lead to hepatic fibrosis [14]. Over time, ROS and chronic low-grade inflammation lead to fibrosis and organ dysfunction [14].

The current treatment for Chagas disease is administering the antiparasitic drug benznidazole (BNZ) [10]. While BNZ is effective at curing when treatment is initiated during the acute phase, cure rates significantly decline when treatment is initiated during the chronic phase [15]. Importantly, a large multicenter trial treating patients with established cardiac disease showed that treatment did not prevent disease progression or cardiac death [16]. Another important limitation of benznidazole therapy is the toxic profile. The side effects of BNZ include hypersensitivity, neuropathy, and bone marrow disorders, which can result in individuals discontinuing treatment [17]. Benznidazole is metabolized by the cytochrome p450 enzyme, which is found throughout the body but primarily in liver cells [18]. The liver has also been shown to efficiently elicit a robust immune response with superior levels of inflammation and IFN-γ production [19]. These elevated levels of inflammation can also damage the liver and may be associated with immune allergic hepatitis [20]. Further, it has been demonstrated in pre-clinical models of acute Chagas disease that the combined effect of both infection and BNZ treatment induce more liver damage than either component alone [21]. Thus, it is critical to evaluate the impact of novel treatments for Chagas disease on the liver to avoid exacerbating damage and diminishing overall health.

To overcome the efficacy and tolerability limitations of standard BNZ treatment, we developed a recombinant protein vaccine, consisting of the *T. cruzi* flagellar derived Tc24-C4 antigen combined with a TLR4 agonist adjuvant in a stable squalene emulsion [22,23]. In mouse models of acute infection, this vaccine effectively reduces parasite burdens, cardiac inflammation, and cardiac fibrosis when combined with low-dose BNZ treatment in a vaccine-linked chemotherapy strategy [24,25]. Further, vaccine-linked chemotherapy improved cardiac function, and reduced endpoint cardiac inflammation in a mouse model of chronic infection similar to curative doses of BNZ [26]. This multimodal approach reduces the dosing requirements of BNZ while showing the ability to enhance the efficacy of the drug and reduce tissue damage by increasing IFN-γ production. Thus, this dose sparing strategy should reduce the tolerability concerns of standard BNZ treatment. However, the impact of vaccine-linked chemotherapy on liver health has not been explored. Therefore, we specifically evaluated the effect of both a curative BNZ treatment regimen as well as our vaccine-linked chemotherapy regimen on liver health in a mouse model of chronic *T. cruzi* infection.

## Material and methods

### Ethics statement

All animal studies were conducted in strict compliance with the 8th Edition of The Guide for Care and Use of Laboratory Animals and were approved by the Baylor College of Medicine Institutional Animal Care and Use Committee under assurance number D16-00475.

## Mice and parasites

Six- to eight-week-old BALB/c female mice were obtained from Taconic (Rensselaer, NY). Mice were housed in groups of 5 in static caging, with ad-libitum food and water, and kept on a 12:12hr light/dark cycle. The *T. cruzi* H1 strain, transfected with the pTRIX2-RE9h plasmid containing the thermostable, red-shifted firefly luciferase gene PpyRE9h [27–30], was grown on monolayers of the C2C12 mouse myocyte cell line (ATCC CRL-1772) in RPMI media supplemented with 5% fetal bovine serum and 1X Penicillin/Streptomycin (cRPMI) to propagate tissue culture trypomastigotes (TCT). Culture media containing TCT was collected, parasites were pelleted by centrifugation, washed once with sterile medical grade saline, then resuspended in sterile medical grade saline. We elected to use a luciferase expressing clone of the *T. cruzi* H1 strain for these studies as we have demonstrated that chronic infection with this clone induces significant cardiac pathology in our female BALB/c mouse model, including changes in cardiac structure and function (30), similar to changes induced by chronic infection with the WT *T. cruzi* H1 strain used in prior studies (26). Use of this clone will allow serial *in vivo* imaging in future studies to track and quantify tissue parasite levels.

## Benznidazole

Benznidazole powder (MedChem Express) was resuspended in 5% DMSO/95% HPMC (0.5% hydroxypropyl methylcellulose/ 0.4% Tween 80/ 0.5% benzyl alcohol in deionized water) to a final concentration of 10mg/mL. Mice were administered a 25mg/kg BNZ or 100mg/kg BNZ (Table 1) as described in the study design.

## Vaccine formulations

Recombinant Tc24-C4 protein was expressed and purified in-house according to previously published protocols [23]. The TLR4 agonist adjuvant E6020 (Eisai, Inc) was dissolved in a stable squalene emulsion (SE). Vaccine formulations comprised of the selected dose of recombinant Tc24-C4 protein and E6020 in 4% squalene emulsion in PBS 1x pH 7.4 were freshly prepared and mixed (1:1) just before subcutaneous injection. Mice were administered 25μg Tc24-C4/ 5μg E6020-SE or 5μg E6020-SE alone (Table 1) as described in the study design.

## Study design

Mice were infected with 5000 trypomastigotes of a bioluminescent clone of the *T. cruzi* H1 strain, generated in our laboratory [30], by intraperitoneal injection. Naïve age-matched mice were left uninfected as controls. Blood was collected by tail vein microsampling from all mice at approximately 28 days post-infection (DPI) to confirm parasitemia by quantitative PCR. Approximately 70 DPI mice were randomly assigned to treatment groups, with 15 mice per group as described in Table 2. Benznidazole treatments were administered once daily by oral gavage, and vaccinations were administered by subcutaneous injection according to the timeline in Fig 1. Mice were monitored daily for morbidity and any mice that reached humane

**Table 1. Treatments, routes, doses and frequency.**

| Treatment | Dose | Route | Frequency |
|---|---|---|---|
| Vaccine | 25μg Tc24-C4/ 5μg E6020-SE | Subcutaneously | Twice, two weeks apart |
| E6020 SE | 5μg E6020-SE | Subcutaneously | Twice, two weeks apart |
| Low BNZ | 25mg/kg | Orally | Once daily for 18 days |
| Curative BNZ | 100mg/kg | Orally | Once daily for 18 days |

**Table 2. Treatment groups.**

| Group | Infection | Treatment #1 | Treatment #2 | Euthanasia Timepoint |
|-------|-----------|--------------|--------------|----------------------|
| #1 | Naïve | N/A | N/A | 142dpi |
| #2 | Infected | N/A | N/A | 90dpi |
| #3 | Infected | N/A | N/A | 120dpi |
| #4 | Infected | N/A | N/A | 142dpi |
| #5 | Infected | Low BNZ | N/A | 120dpi |
| #6 | Infected | Vaccine | N/A | 120dpi |
| #7 | Infected | Low BNZ | Vaccine | 142dpi |
| #8 | Infected | E6020 SE | N/A | 120dpi |
| #9 | Infected | High BNZ | N/A | 142dpi |

endpoints were humanely euthanized. Cohorts of mice were euthanized at multiple time points after treatments were completed to evaluate the treatment's short- and long-term effects on liver health. At the study endpoints, all mice were weighed, then humanely euthanized. Whole blood was collected postmortem. Livers were removed, and weighed, then one section was frozen for DNA and RNA analysis, and a second portion was placed in 10% neutral buffered formalin for histopathology analysis.

## QRT-PCR for Parasite Burden

According to the manufacturer's guidelines, DNA was extracted from frozen liver tissue (20mg) using PDQeX Nucleic Acid Extractor. To measure tissue parasite burden, quantitative real-time PCR was performed using TaqMan Fast Advanced master mix (Life Technologies) and oligonucleotides specific for the satellite region of *T. cruzi* nuclear DNA (primers 5′–ASTCGGCTGATCGTTTTCGA–3′ and 5′–AATTCCTCCAAGCAGCGGATA–3′ and probe 5′–6–FAM–CACACACTGGACACCAA–MGB–3′. *T. cruzi* data were normalized to glyceraldehyde-3-phosphate dehydrogenase (GAPDH) (primers 5′ CAATGTGTCCGTCGTGGATCT 3′

**Experimental study design**

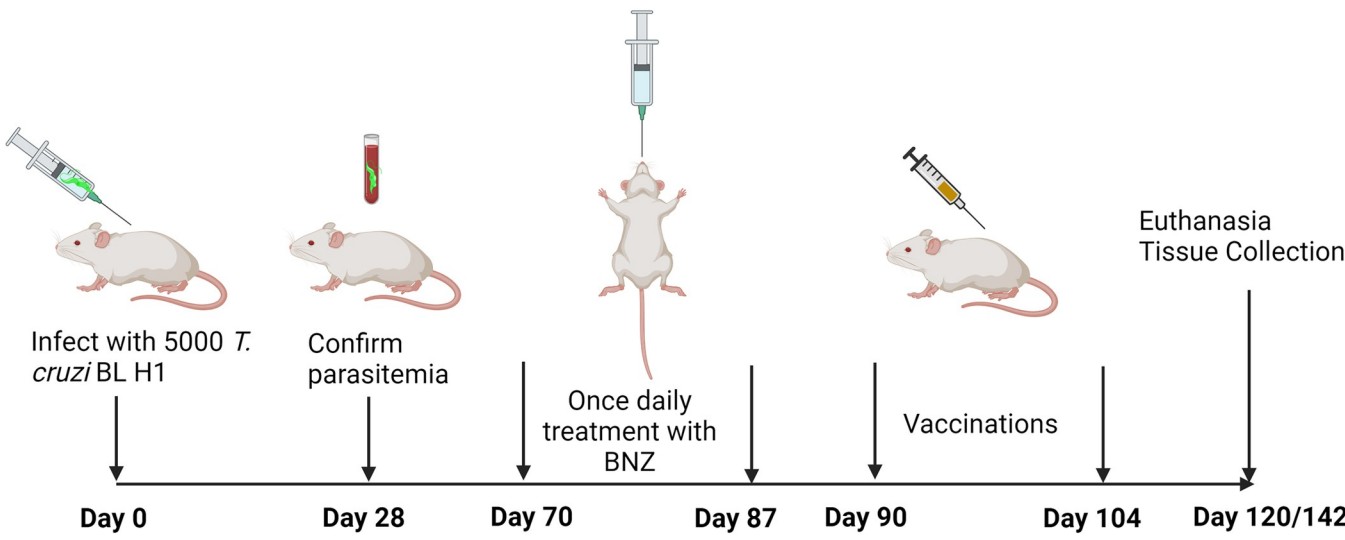

**Fig 1. Treatment timeline.** Image created with Biorender.

and 5' `GTCCTCAGTGTAGCCCAAGATG` 3', probe 5' `6-FAM CGTGCCGCCTGGA GAAACCTGCC MGB` 3'; Life Technologies, CA, USA) (Life Technologies), and parasite burden was calculated based on the standard curve of known parasite contents [24].

### Liver enzyme analysis

Whole blood was allowed to clot at room temperature for 30 minutes, then centrifuged at 10,000 rpm for 5 minutes at room temperature to separate serum. Serum was analyzed for aspartate aminotransferase (AST), alanine aminotransferase (ALT) and alkaline phosphatase (ALP), and lactate dehydrogenase (LDH) using a Beckman Coulter AU480 Chemistry Analyzer (Clinical Pathology Laboratory, Baylor College of Medicine).

### Histopathology analysis

Sections of the liver were fixed in 10% neutral buffered formalin, dehydrated, embedded in paraffin, sectioned to 5μm thickness, and adhered to glass slides. Sections were routinely stained with hematoxylin and eosin (H&E). Images of liver H&E histology slides were taken from 5 randomly selected representative fields at 10x magnification using an Amscope ME580 brightfield microscope. Images' color thresholds (hue, saturation, brightness) were adjusted using ImageJ to create uniformity among all images. A count of lymphocyte nuclei was accomplished by setting a particle size limit to exclude larger hepatocyte nuclei and smaller debris from being factored into the count. Lymphocyte counts from all 5 randomly selected fields were averaged to indicate inflammatory cell infiltration in the liver.

### QRT-PCR for BTG2 and PPARα

According to the manufacturer's guidelines, RNA was isolated from frozen liver tissue (20mg) using RNeasy kit (Qiagen). The concentration of RNA was quantified using Nanodrop with a target concentration of 100ng/uL. cDNA was amplified with RT-PCR master mix (Thermo-Fisher) and ran in Bio-Rad PCR Thermal Cycler. QRT-PCR was performed using a Quant Studio 3 thermocycler (Applied Biosciences). The specific primers were as follows: BTG2 (Mm00476162_m1 Primers and Probe Taqman gene Btg2 (FAM-MGB), PPARα (Mm00440939_m1) Primers and Probe Taqman gene Pparα (FAM-MGB) (Applied Biosciences). All samples were run in duplicate. The relative quantity (RQ) values were calculated according to the ΔΔCt method. The infected mice were normalized to the values obtained from non-infected mice (ΔΔCT). Then, the RQ was calculated as RQ = 2-ΔΔCt.

### Statistical analysis

For each parameter measured, data were plotted using GraphPad Prism 9.4.1 software (Graph-Pad). Treatments were compared at each timepoint to infected untreated controls or naïve mice, as indicated in the figures using a Kruskal-Wallis one-way ANOVA and Dunn's multiple comparisons tests. When comparing only two groups, a Mann-Whitney test was used. P values $\leq 0.05$ were considered significant.

## Results

### Curative Benznidazole treatment reduces *T. cruzi* induced hepatomegaly

In Liu et al, 2023, we previously reported that combination treatment with low BNZ + vaccine better restored *T. cruzi* induced metabolic perturbances in several sections of heart compared to curative BNZ treatment [30]. However, that work did not evaluate the impact of treatments on liver health. Hepatomegaly is a consistent finding in human cases of Chagas disease as well

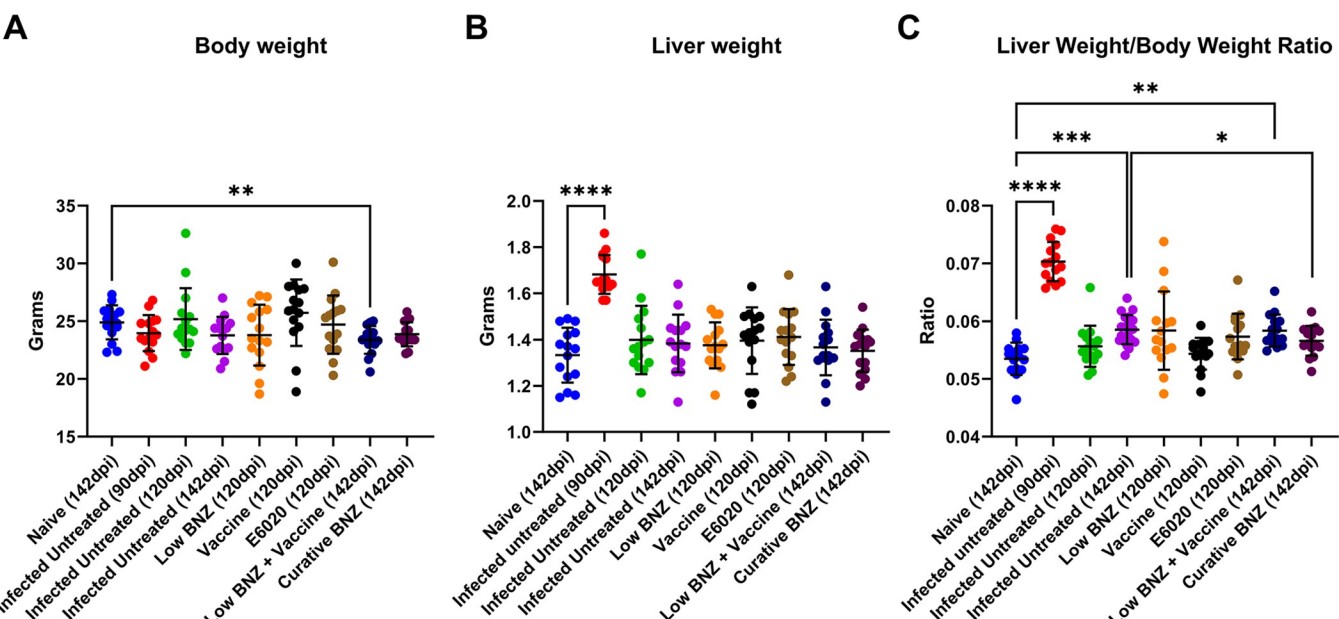

**Fig 2. Body weight, liver weight, and liver weight/body weight ratio of mice at 90 dpi, 120dpi and 142dpi.** Body (**A**) and liver (**B**) weights were taken at time of euthanasia. A ratio was calculated to normalize results. *$P < 0.05$; ***$P < 0.0005$; ****$P < 0.0001$.

as experimental animal models [31]. To determine if hepatomegaly was also present in our model of chronic *T. cruzi* infection, the liver weight/ body weight ratio was calculated at study endpoints. The liver weight/ body weight ratio was significantly increased by infection at 90 and 142 DPI (Fig 2C, red and purple symbols, respectively) compared to naïve controls, indicating hepatomegaly was evident in our model. By 142 DPI, curative BNZ significantly reduced liver weight/body weight ratio compared to infected control mice (Fig 2C maroon symbol), suggesting that curative BNZ treatment ameliorates infection-induced hepatomegaly. However, low BNZ + vaccine had no apparent effect on hepatomegaly. Additionally, infection alone did not induce significant changes in overall body weight compared to naïve controls, but low BNZ + vaccine resulted in overall lower body weight compared to controls (Fig 2A). Overall liver weight was significantly increased only at 90 DPI when compared to naïve controls, but liver weight was not increased at other timepoints or with any treatments. Together, these data confirm hepatomegaly is evident in our mouse model and that curative BNZ reduces infection induced hepatomegaly by 142dpi, but does not restore liver weight/body weight ratio to the same level as age matched naïve mice.

### Curative Benznidazole clears liver parasites and reduces cellular infiltration

We have previously demonstrated that vaccine-linked chemotherapy significantly reduces cardiac parasite burdens in acutely infected mice [24,25] and curative BNZ significantly reduces cardiac parasite burdens and reduces cellular infiltration in chronically infected mice immediately after treatment [32]. Therefore, we evaluated the impact of treatments on parasite levels and cellular infiltration in the liver. Infection resulted in significantly increased infiltration of inflammatory cells into the liver at 90dpi and 142 dpi compared to naïve mice at 142dpi (Fig 3B red and purple symbols, respectively). Additionally, infection induced inflammatory infiltrate was significantly reduced at 120dpi compared to 90 dpi (Fig 3B green and red symbols, respectively), but inflammatory infiltrate at 142dpi was not significantly different to 90dpi (Fig 3B maroon and red symbols respectively). Curative BNZ effectively decreased parasite burden

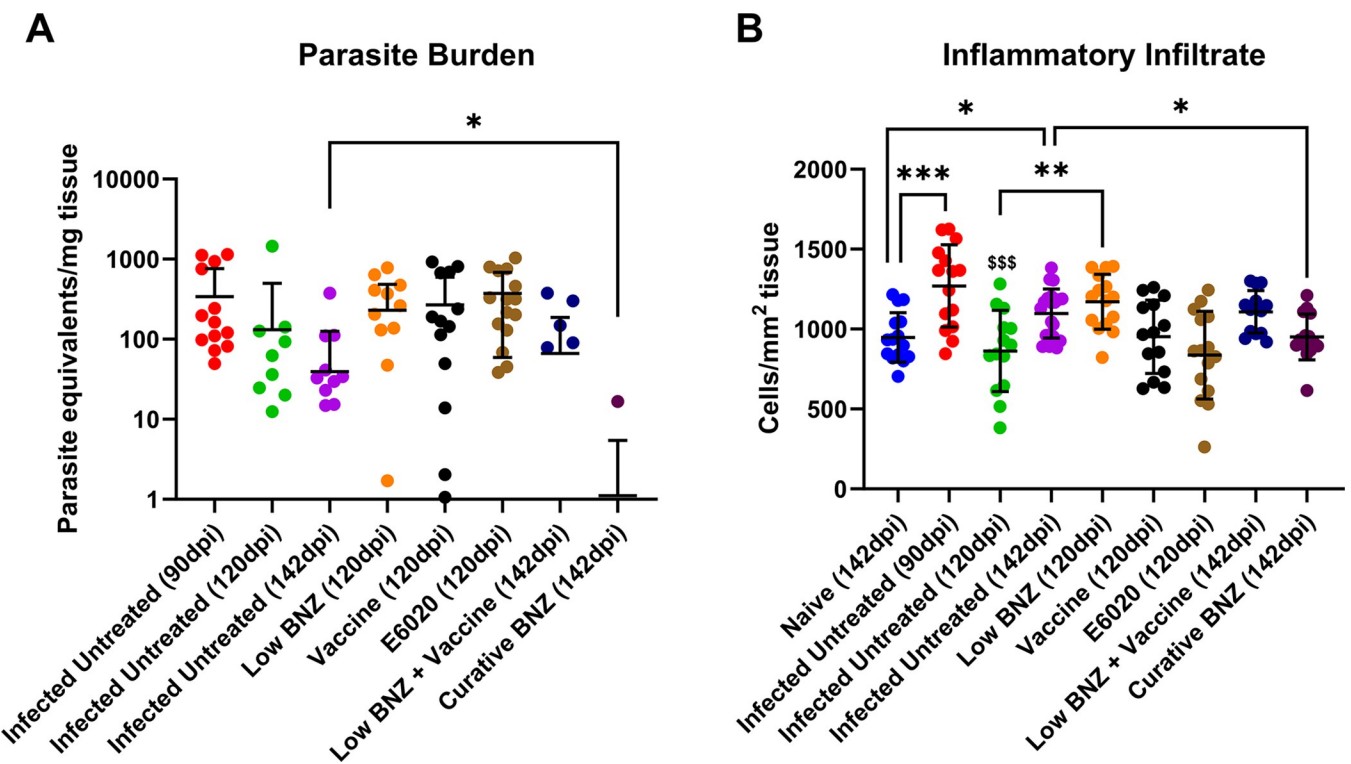

**Fig 3. Parasite burden and amount of inflammatory infiltrate in the liver.** *T. cruzi* parasite burden expressed as parasite equivalents per mg of tissue (**A**) was determined using qPCR and inflammatory infiltrate was determined via histopathology analysis using ImageJ (**B**). *$P < 0.05$; **$P < 0.005$. \$ \$ \$ $P < 0.001$ when compared to Infected Untreated at 90dpi.

in the liver to below the limits of quantitation for the assay (Fig 3A maroon symbol), while low BNZ + vaccine had no effect on parasite burdens (Fig 3A blue symbols). Evaluation of H&E stained liver sections revealed that treatment with low BNZ significantly increased inflammatory infiltrate at 120 DPI compared to infected untreated mice. (Fig 3B orange symbol Fig 4E). However, curative BNZ significantly decreased inflammatory infiltrate compared to infected untreated mice (Fig 3B maroon symbol, Fig 4I), while low BNZ + vaccine did not affect the number of inflammatory cells (Fig 3B dark blue symbol, Fig 4H).

### Curative Benznidazole significantly elevates serum tissue damage markers

To determine the effect of treatments on tissue damage markers as indicators of liver damage, we evaluated levels of ALT, ALP, AST and LDH. By 142dpi, both curative BNZ and low BNZ + vaccine induced significant elevations to ALT (Fig 5A, navy and maroon symbols, respectively) and AST (Fig 5C, navy and maroon symbols, respectively) and AST when compared to naïve mice. Importantly, only curative BNZ induced significant elevation to LDH (Fig 5D, maroon symbols) compared to naïve mice. Infection alone and single treatments did not cause significant elevations to AST, ALT and LDH. Further, no differences in ALP were observed for any groups. Together, these data suggest that our vaccine-linked chemotherapy strategy causes less liver and tissue damage compared to curative BNZ alone.

### Benznidazole treatment reduces the expression of liver damage markers

To begin to define potential mechanism of liver damage in our model, we evaluated expression of BTG2, a marker of oxidative damage, and PPARα, a regulator of inflammation [33–35].

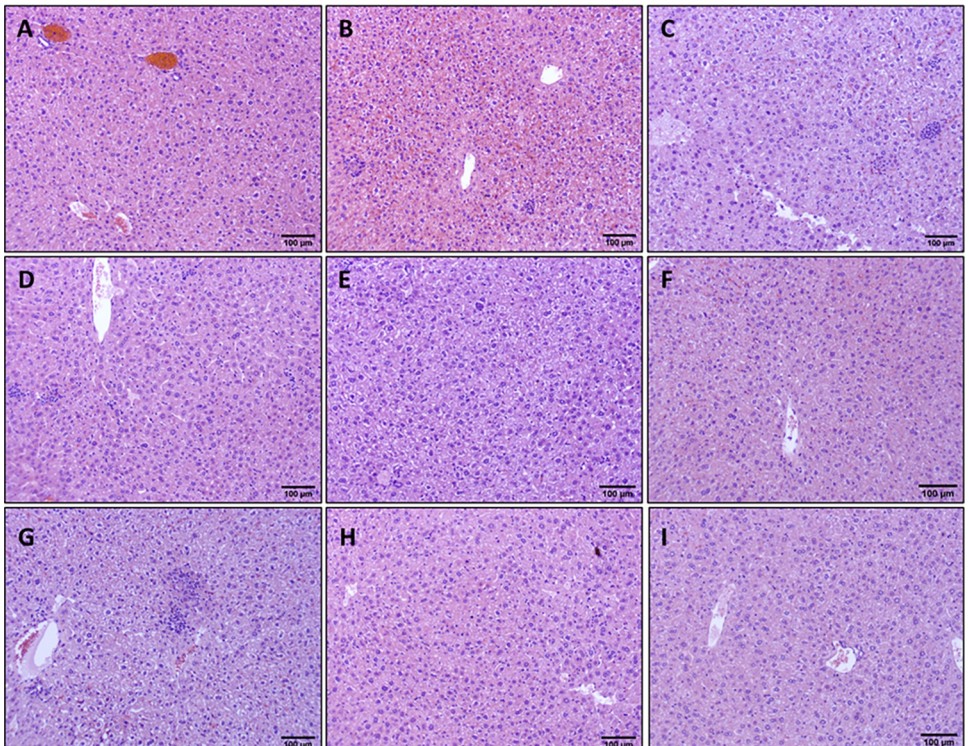

**Fig 4. Impact of therapeutic interventions on the inflammatory infiltrate of liver tissue in mice.** Representative H&E stained sections are shown at 142dpi Naïve (A), 90dpi Infected Untreated (B), 120 Infected Untreated (C), 142dpi Untreated (D), 120dpi Low BNZ (E), 120dpi Vaccine (F), 120dpi E6020 (G), 142dpi Vaccine + Low BNZ (H), 142dpi Curative BNZ (I). The scale bar represents 100um.

BTG2 expression was elevated by 120 DPI in infected mice compared to 90 DPI and 142DPI (Fig 6A green symbols), and expression at 142dpi was significantly elevated compared to 90 DPI (Fig 6A purple and red symbols respectively). BTG2 expression was significantly decreased by low BNZ treatment (Fig 6A orange symbols) compared to infection alone at 120 DPI (Fig 6A green symbols). Similarly, by 142 DPI, infection induced BTG2 expression was significantly reduced by curative BNZ treatment (Fig 6A maroon symbols). Expression of PPARα was significantly elevated by infection at 142 DPI (Fig 6B purple symbols) compared to 90DPI and 120DPI (Fig 6B red and green symbols, respectively). Only low BNZ significantly decreased PPARα expression by 120DPI (Fig 6B orange symbols) compared to infected mice at 120 DPI (Fig 6B orange symbols). Together, these data suggest that while low dose BNZ treatment can improve oxidative damage and regulation of inflammation in the liver, the combination of low BNZ + vaccine does not result in similar improvement.

## Discussion

The current first-line treatment for Chagas disease is oral BNZ, which is effective mainly in the acute phase but has unwanted side effects, including dermatologic and neurologic manifestations [36,37]. Liver toxicity from BZN is thought to be caused by the formation of reactive metabolites, which can damage liver cells and lead to inflammation and liver injury [36]. Symptoms of BZN-induced liver toxicity may include fatigue, abdominal pain, jaundice, and elevated liver enzymes in the blood [20]. Specifically, BNZ treatment results in elevations of AST, ALT, and ALP [38]. Our group has shown that a vaccine-linked chemotherapy strategy

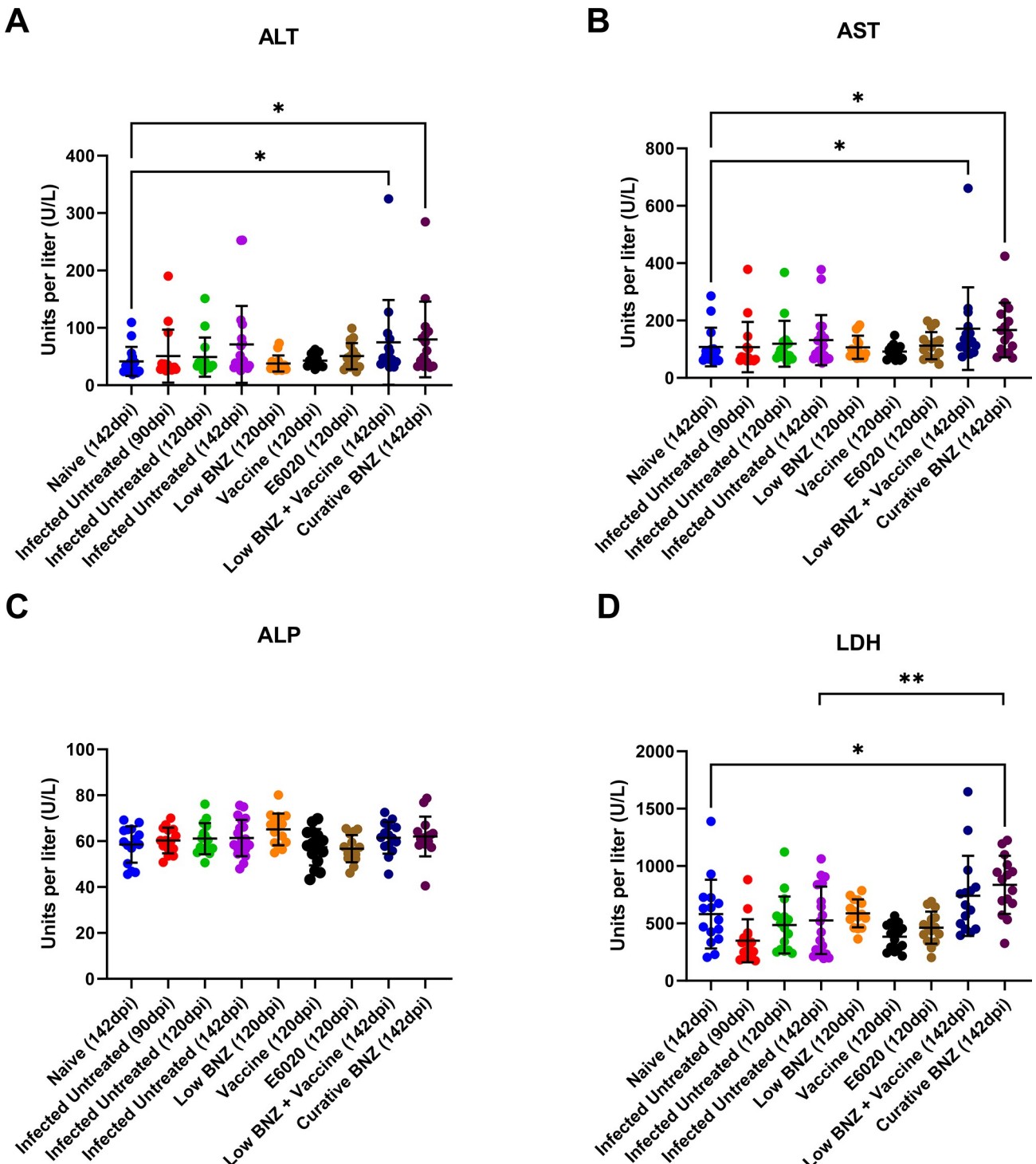

**Fig 5. Impact of infection and treatment on liver enzymes.** Complete serum chemistry analysis was performed. ALT, ALP, AST, and LDH were evaluated because they are the most common liver function tests. *P ≤0.05; **P ≤0.01.

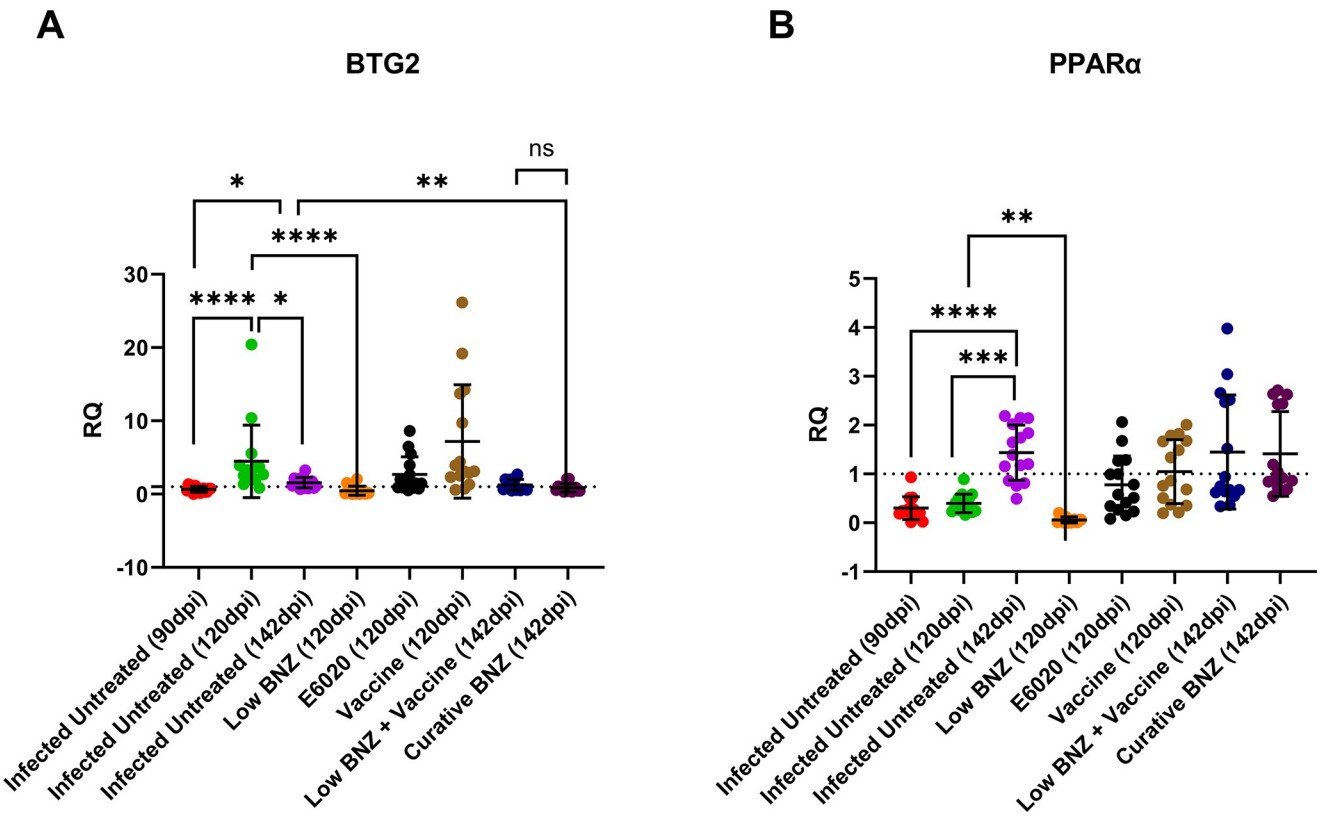

**Fig 6. Infection status and treatments on expression of BTG2 and PPARα.** BTG2 and PPARα were measured in liver tissue. *P<0.05; **P < 0.01; ***P<0.001 ****P < 0.0001.

allows the reduction of BNZ dose, while still effectively reducing cardiac inflammation and fibrosis, and improving cardiac function [24–26]. While the evidence showed that this dose-sparing strategy improved cardiac health, similar to a curative dose of BNZ, the impact of vaccine-linked chemotherapy on liver health was unknown. Therefore, we conducted this study to compare the effects of curative benznidazole treatment and vaccine-linked chemotherapy on liver health. *T. cruzi* infection causes inflammation and edema, which can contribute to hepatomegaly [31]. Additionally, right-sided heart failure can cause congestion of the liver; thus, an increased liver weight can indicate cardiac dysfunction [39,40]. We confirmed in our model that curative BNZ treatment was able to ameliorate the infection-induced hepatomegaly, decrease liver parasite burdens, and reduce cellular infiltrate. This agrees with our prior observations in our chronic infection mouse model that curative BNZ significantly reduced cardiac parasite burdens and cardiac cellular infiltration immediately after treatment was completed [32], and reduced cardiac cellular infiltration over 3 months after treatment was completed [26]. In contrast, low BNZ + vaccine had no apparent effect on *T. cruzi* induced hepatomegaly, liver parasite burdens, or cellular infiltrate. We have previously shown that our Tc24-C4/E6020 SE vaccine induced increased antigen specific CD8+ cells in the spleen [25], thus it is possible that the cellular infiltrate within the liver of vaccinated mice showed no apparent reduction due to increased infiltration of antigen specific CD8+ cells into tissues, which could also contribute to overall enlargement. Additionally, in a dog model of acute *T. cruzi* infection, preventative vaccination with a DNA vaccine modified the composition of the cardiac inflammatory infiltrate from primarily mononuclear cells in infected unvaccinated

dogs, to polymorphonuclear cells in infected vaccinated dogs [41]. Further analysis of the specific cell types present in the liver would be necessary to define specific cell types in inflammatory infiltrate and determine if low BNZ + vaccine modified the composition of cells. Additionally, while we did not observe reductions in parasite burdens in the liver in mice treated with low BNZ + vaccine, it is possible that parasite burdens were reduced in other organs. Evaluation of other tissues including gastrointestinal tract, which is a demonstrated reservoir for the parasite [29], would be necessary to determine if treatment reduced overall parasite burdens.

Despite the positive impact of curative BNZ on hepatomegaly, parasite burden, and inflammation, significant elevations in AST, ALT and LDH indicate that tissue damage was still evident. ALT is abundantly expressed in the liver, and elevated serum levels indicate liver damage, while AST and ALP are expressed in multiple tissues, including the liver, heart, and muscle [42–44]. In experimental mouse models of acute Chagas disease, it has been demonstrated that the combined effect of both infection and BNZ treatment induced more elevation of serum AST, ALT, and ALP, as well as direct liver damage on histopathology, than either infection or BNZ administration alone [21]. Our results showed elevated serum ALT and AST levels in the group treated with the curative dose of BNZ, indicating that in mouse models of chronic Chagas disease curative BNZ treatment causes elevations in tissue damage markers similar to acute infection models. In our study, curative BNZ treatment was administered from approximately 70 dpi until 87 DPI; thus, endpoint serum samples collected at 142 DPI were 55 days after treatment ended. Since both AST and ALT were still elevated at that time in mice treated with curative BNZ, this indicates that despite significantly reducing parasite burdens and inflammatory cell infiltrate curative BNZ treatment caused sustained tissue damage, further supporting the need for less toxic treatment options. Interestingly, treatment with either a low dose of BNZ alone or the Tc24-C4/ E6020 SE vaccine alone did not cause significantly elevated ALT or AST by 120dpi, which was approximately 32 and 36 days after treatment ended, respectively. This suggests low dose BNZ or vaccine do not cause as much tissue damage as curative BNZ treatment. However, the group treated with low BNZ + vaccine sequentially had elevated serum levels of ALT and AST at 142dpi, similar to curative BNZ. This suggests that despite the dose sparing effect of this strategy, the combination of treatments does still result in elevation of tissue damage enzymes, possibly due to combined effects of reactive metabolites resulting from BNZ treatment [36] and activation of inflammatory pathways by E6020, the TLR4 agonist adjuvant use in the vaccine component [45,46]. Further studies evaluating ALT and AST at later timepoints after treatment would be needed to determine if the elevations in these tissue damage markers is sustained or if the elevations are transient and ultimately return to normal levels.

In addition to ALT, AST, and ALP, we evaluated serum lactate dehydrogenase (LDH) levels, which are also abundantly expressed in the liver, heart, and muscle tissues [47,48]. In damaged tissues, LDH leaks out of the tissues and into the serum [49]. The elevated LDH levels of curative BNZ-treated mice suggest that the drug has hepatotoxic effects on tissues, which may lead to acute liver failure [50]. Furthermore, previous studies have also concluded that LDH is a good prognosticator for death in individuals with acute liver failure [51]. Studies in mice acutely infected with *T. cruzi* show that LDH levels are detectable in both serum and tissues at the early stages of infection, with elevations evident in serum before tissue damage is evident microscopically [47]. This suggests that LDH elevations caused by *T. cruzi* are due to the combined effect of early changes in cell membrane permeability and later structural damage to tissues [47]. Our results showed that only treatment with the curative dose of BNZ resulted in significant elevations in serum LDH levels. Importantly, the lack of LDH elevation in mice receiving low BNZ + vaccine suggests that this treatment is less damaging to tissues, potentially

in both the liver and heart, than curative BNZ. Considering the know exacerbation of liver damage caused by the combined effects of *T. cruzi* infection and BNZ [21], a less damaging treatment strategy, such as vaccine-linked chemotherapy, is desirable to preserve overall health.

In an effort to further characterize potential mechanisms of liver-specific damage in our model, we evaluated gene expression of B-cell translocation gene 2 (BTG2) and Peroxisome proliferator-activated receptor alpha (PPARα). BTG2 is a protein-coding gene involved in various cellular processes, including cell cycle regulation, apoptosis, differentiation, hepatic gluconeogenesis, and lipid homeostasis [52–54]. Importantly, BTG2 is induced in response to DNA damage, and upregulation of BTG2 has been shown to protect against oxidative stress in human mammary epithelial cells [33,34]. *T. cruzi* infection leads to DNA damage in mouse models, specifically in heart cells and splenocytes, due to induction of reactive oxygen species (ROS), and BNZ has been shown to reduce that effect [55,56]. In our study, treatment with either low BNZ or curative BNZ significantly decreased BTG2 expression levels compared to infected, untreated mice at 120 DPI and 142 DPI, respectively. This suggests that in addition to DNA damage in the heart, *T. cruzi* also induces DNA damage in the liver leading to elevated BTG2 expression, which is ameliorated with BNZ treatment. BTG2 expression is also upregulated in response to inflammatory stimuli, including IL-6 and NFκB [57]. In prior studies, we showed that curative BNZ treatment of chronically infected mice did not significantly reduce NFκB, pSTAT3, or IL-6 in cardiac tissue [32]. However, specific evaluation of NFκB and IL-6 in the liver would be needed to determine if curative BNZ treatments has a tissue specific impact on those inflammatory stimuli. PPARα is involved in regulating lipid and glucose metabolism, is abundantly expressed in the liver, and is critical for reducing inflammation and protecting against liver injury [35,58,59]. In *T. cruzi* infected macrophages, PPARα ligands drive M1-to-M2 conversion, regulating inflammatory responses [60]. We observed that low BNZ significantly reduced expression of PPARα in the liver, similar to the effect on BTG2 expression. While this reduction did not have an apparent effect on liver cellular infiltrate, further studies would be needed to determine any impact on other inflammatory markers specifically in the liver.

Limitations of this study were identified and have been considered in the interpretation of our results and planning of future studies. This study focused on evaluating liver damage in our female mouse model of *T. cruzi* infection, which we have used extensively to confirm efficacy of our vaccine-linked chemotherapy strategy [24–26]. Additionally, studies of sex related differences in drug induced liver toxicity have demonstrated that women are more likely to present with drug-induced hepatotoxicity [61]. However, a 10 year longitudinal study of patients with Chagas disease found that male patients developed cardiac disease at a higher rate compared to female patients [62]. Further, heart failure can also result in liver damage [63]. Thus, in future studies it will be essential to specifically evaluate the impact of *T. cruzi* infection and vaccine-linked chemotherapy on liver health in male mice and compare to the impact in female mice. Another limitation identified in this study is the limited time points after infection and treatment that were evaluated. In this study we evaluated liver health primarily at 120 DPI and 142 DPI, representing early chronic infection, to evaluate short term effects of our treatments compared to infected untreated mice at each timepoint. However, future studies will need to incorporate evaluation of all groups immediately after treatment as well as at later time points to determine the duration of any treatment effects on liver health. Indeed, we observed elevated AST and ALT in mice treated with low BNZ + vaccine at 142DPI, representing approximately 36 days after treatment ended. It is possible that at later timepoints, AST and ALT levels would return to normal range indicating that any toxic effects in the liver are transient. Finally, cardiac fibrosis is a consistent finding in chronic Chagasic

cardiomyopathy, and is associated with more severe cardiac disease [64,65]. Because the relationship between the heart health and liver health has been demonstrated, assessing whether our treatment affects liver fibrosis will also be important.

BNZ treatment of patients with Chagas disease is problematic due to prolonged treatment courses and significant toxicity, resulting in up to 40% of patients terminating treatment early [15,37]. We developed a vaccine-linked chemotherapy strategy that is dose-sparing, and demonstrated to be efficacious at reducing cardiac pathology in preclinical models [24–26]. Here we present data suggesting that in addition to the beneficial cardiac effects, vaccine-linked chemotherapy may be less damaging to the liver compared to curative BNZ treatment. While additional studies are needed to develop optimized multi-modal treatment strategies that further improve liver health, these data further support vaccine-linked chemotherapy as an attractive strategy to bridge the efficacy and tolerability gaps of standard anti-parasitic treatment for patients with Chagas disease, and ultimately improve overall health.

## Acknowledgments

This work was supported by the Center for Comparative Medicine Research Services Laboratory at Baylor College of Medicine.

## Author Contributions

**Conceptualization:** Maria Elena Bottazzi, Peter J. Hotez, Kathryn Marie Jones.

**Formal analysis:** Duc Minh Nguyen, Cristina Poveda, Kathryn Marie Jones.

**Funding acquisition:** Peter J. Hotez.

**Investigation:** Duc Minh Nguyen, Cristina Poveda, Kathryn Marie Jones.

**Methodology:** Maria Elena Bottazzi, Peter J. Hotez, Kathryn Marie Jones.

**Resources:** Jeroen Pollet, Fabian Gusovsky.

**Supervision:** Maria Elena Bottazzi, Peter J. Hotez, Kathryn Marie Jones.

**Visualization:** Duc Minh Nguyen, Cristina Poveda, Kathryn Marie Jones.

**Writing – original draft:** Duc Minh Nguyen, Cristina Poveda, Kathryn Marie Jones.

**Writing – review & editing:** Duc Minh Nguyen, Cristina Poveda, Kathryn Marie Jones.

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
