## [Decision Letter · Decision Letter 0]

10 Oct 2023

Dear Kathryn Jones,

Thank you very much for submitting your manuscript "The impact of vaccine-linked chemotherapy on liver health in a mouse model of chronic Trypanosoma cruzi infection" for consideration at PLOS Neglected Tropical Diseases. As with all papers reviewed by the journal, your manuscript was reviewed by members of the editorial board and by several independent reviewers. The reviewers appreciated the attention to an important topic. Based on the reviews, we are likely to accept this manuscript for publication, providing that you modify the manuscript according to the review recommendations. 

Sincerely,

Renata Rosito Tonelli, PhD

Academic Editor

Abhay Satoskar

Section Editor

Reviewer's Responses to Questions

**Key Review Criteria Required for Acceptance?**

**Methods**

-Are the objectives of the study clearly articulated with a clear testable hypothesis stated?

-Is the study design appropriate to address the stated objectives?

-Is the population clearly described and appropriate for the hypothesis being tested?

-Is the sample size sufficient to ensure adequate power to address the hypothesis being tested?

-Were correct statistical analysis used to support conclusions?

-Are there concerns about ethical or regulatory requirements being met?

Reviewer #1: No issues

Reviewer #2: This study was designed to verified weather low doses of the anti-parasitic drug benznidazole (BNZ) combined with a prototype therapeutic vaccine is effective in controlling parasite infection without causing liver damage. BNZ is a drug known to cause liver damage if administered with a curative dose. In a previous study the authors showed that a combination of low dose plus a recombinant protein vaccine can effectively reduce T. cruzi induced cardiac damage. Here, the authors evaluated several markers of liver health after treatment with low dose BNZ plus the vaccine therapy. The authors showed that, as expected, treatment of infected mice with curative doses of BNZ reduces parasite burden but elevates serum levels of several toxicity markers. However, neither low doses of BNZ or low doses plus vaccine results in reduced parasite burden but low doses plus vaccine results in a slight better outcome in terms of liver toxicity compared to the effect of a curative dose of BNZ. This is a well conducted and useful study that may help optimize the treatment for Chagas disease. All the objectives of the study were clearly articulated with a testable hypothesis, and the study design appropriately addresses its objectives. Proper statistical analysis was used to support the conclusions and all ethical requirements regarding animal use have been met.

**Results**

-Does the analysis presented match the analysis plan?

-Are the results clearly and completely presented?

-Are the figures (Tables, Images) of sufficient quality for clarity?

Reviewer #1: See general comments

Reviewer #2: The experiments were well designed, and the results of the analysis were clearly presented, with all figures having good quality. As indicated below, I have few criticisms regarding the lack of statistical significance analyses of part of the data, which would be important to strengthen some of the authors conclusions. 

Line 160, the authors used bioluminescent parasites but on line 180 they indicated that tissue parasite burden was determined only by quantitative real-time PCR. Please explain why luciferase expressing parasites were used. Also, regarding parasite burden, on Figure 3 it should be indicated how the parasite numbers were calculated (is it per mg of tissue?).

Fig 2, line 238, the authors claimed that “these data confirm hepatomegaly is evident in our model and that only curative BNZ ameliorates this finding.” However, the difference in the liver weight/body weight is minimal. Therefore, the sentence should be changed to “these data confirm hepatomegaly is evident in our model and that curative BNZ slightly ameliorates this finding”. None of the other treatments had an impact in liver health based on the liver weight/body weight ratio. Why the liver weight/body weight ratio was not measured in animals treated with low plus vaccine and curative BNZ doses at 90 dpi, when this ration was much higher compared to the ration at 142 dpi?

Fig 3, lane 254: are there statistically significance differences in inflammatory infiltrate between infected untreated at 90 dpi and 120 dpi? Or between naïve and infected untreated at 142 dpi? Please indicate in the text. Again, like the liver weight/body weight ratio, the main increase in inflammatory infiltrate is observed at 90 dpi. Why was the effect of curative BNZ only measured 142 dpi when just a small decrease compared to infected untreated mice was observed? Also, the authors claims that curative BNZ significantly decreased inflammatory infiltrate compared to infected untreated mice, however, it was not clear whether the inflammatory infiltrate in the liver of untreated infected mice was increased compared to naïve mice. 

Fig 5, lane 273: are there statistically significance differences in levels of ALT and AST between naïve and infected untreated mice at any dpi? If not, the observations that both curative BNZ and low BNZ + vaccine induced significant elevations to ALT by 142dpi, compared to naïve animals may suggests that the liver damage occurs in response to the treatment and not to the infection. Increased tissue damage was detected when infected and curative BNZ treated mice was compared to mice infected at the same dpi. Based solely on this, the authors could not claim that the “vaccine-linked chemotherapy strategy causes less liver and tissue damage compared to curative BNZ alone” 

Fig 6, line 289: In contrast to the results obtained with enzyme assays, it looks like the authors were able to observe signs of increased liver damage in infected untreated animals at 120 dpi compared to 90 dpi and when 142 dpi was compared to 90 dpi using the expression of a marker of oxidative damage (BTG2) and a regulator of inflammation (PPARα). Again, are these differences statistically significant? If so, it should be mentioned. It should be also mentioned that although low BNZ treatment can reduce oxidative damage and regulation of inflammation, low BNZ + vaccine does not contribute for a reduction of these parameters the same way that this treatment does not cause a reduction in inflammatory infiltrate.

**Conclusions**

-Are the conclusions supported by the data presented?

-Are the limitations of analysis clearly described?

-Do the authors discuss how these data can be helpful to advance our understanding of the topic under study?

-Is public health relevance addressed?

Reviewer #1: See general comments

Reviewer #2: Most of the authors conclusions are supported by the data presented, but, for the sake of clarity, in the abstract, the conclusion sentence should be modified to “These data confirm toxicity associated with curative doses of BNZ and suggest that, although the dose sparing low BNZ plus vaccine treatment does not reduce parasite burden, it better preserves liver health.”

**Editorial and Data Presentation Modifications?**

Reviewer #1: See general comments

Reviewer #2: no comments

**Summary and General Comments**

Reviewer #1: This paper focuses on liver toxicity issues during Chagas disease and during therapy using a mouse model. Parasite infected animals are compared with infected animals treated with either high-dose benznidazole (BNZ) therapy and those treated with low-dose BNZ+vaccine, a previously described subunit vaccine. This latter combination has been shown to result in decrease cardiac disease while have the advantage of reduced exposure to the toxicities of high dose BNZ monotherapy. Overall, the study found that high dose BNZ reduced liver parasite burden and inflammation but leads to liver toxicity as measured liver and tissues enzyme elevations. Low-dose BNZ+vaccine dose not reduce parasite burdens or tissue inflammation but results in similar toxicity to HD-BNZ whereas LD-BNZ alone or vaccine alone does not. 

Overall the experiments are performed well and the data justifies the conclusions. 

1) The major issue is that while the LD BNZ-vaccine combo appears to not have significant liver toxicity, it does not have any impact on parasites in the liver. And while the use of this treatment may indeed be important on cardiac disease it does not treat Chagas liver disease. Whether it reduces or ameliorates parasitization of other tissues is unknown. By extension, how does one treat Chagas liver disease without high dose BNZ? If using LD-BNZ+vaccine for cardiac disease, one would still need another agent to treat disease in the liver or other organs.

2) Both HD-BNZ and LD-BNZ+vaccine treatment induces ALT, AST release from liver of infected mice when compared to naive controls (Fig. 5 A and B). There are no controls of naïve mice treated with either drug alone. In the case of HD-BNZ this could be interpreted as turnover of parasites in the liver affecting hepatocyte toxicity. In the case of LD-BNZ+vaccine there is no demonstrable effect on parasite turnover yet there is still liver toxicity. Why do the authors conclude that this treatment doesn’t produce liver toxicity (lines 279-280 and in line 50), but later state that this isn’t the case (356-359)? LD-BNZ alone or vaccine alone did not result in AST/ALT elevations the combo did leading one to conclude that these synergize to induce toxicity during dual therapy. Perhaps the authors should elaborate on how this may happen. 

3) Regarding the statistics, it is hard to understand why some groups are not statistically different than others just based on the visual representation. For example, Fig 5A, LD-BNZ alone and vaccine alone look even more statistically different to either LD-BNZ+vaccine or HD-BNZ than does naïve mice and LD-BNZ+vaccine or HD-BNZ. Why are they not? 

4) There are some minor grammatical issues: line 357, 375, different fonts between lines 443-445.

Reviewer #2: please see comments above.

PLOS authors have the option to publish the peer review history of their article (what does this mean?). If published, this will include your full peer review and any attached files.

Reviewer #1: No

Reviewer #2: No

Figure Files:

Data Requirements:

Reproducibility:

References

---

## [Editor Report · Decision Letter 1]

9 Nov 2023

Dear Dr Kathryn M. Jones 

We are pleased to inform you that your manuscript 'The impact of vaccine-linked chemotherapy on liver health in a mouse model of chronic Trypanosoma cruzi infection' has been provisionally accepted for publication in PLOS Neglected Tropical Diseases.

Best regards,

Renata Rosito Tonelli, PhD

Academic Editor

Abhay Satoskar

Section Editor

---

## [Editor Report · Acceptance letter]

15 Nov 2023

Dear Dr. Jones,

We are delighted to inform you that your manuscript, "The impact of vaccine-linked chemotherapy on liver health in a mouse model of chronic *Trypanosoma cruzi* infection," has been formally accepted for publication in PLOS Neglected Tropical Diseases.

Best regards,

Shaden Kamhawi

co-Editor-in-Chief

Paul Brindley

co-Editor-in-Chief
